# Chemically Mediated Microbial “Gardening” Capacity of a Seaweed Holobiont Is Dynamic

**DOI:** 10.3390/microorganisms8121893

**Published:** 2020-11-30

**Authors:** Mahasweta Saha, Shawn Dove, Florian Weinberger

**Affiliations:** 1Benthic Ecology, GEOMAR Helmholtz Centre for Ocean Research, Düsternbrooker Weg 20, 24105 Kiel, Germany; fweinberger@geomar.de; 2Marine Ecology and Biodiversity, Plymouth Marine Laboratory, Prospect Place, Plymouth PL1 3DH, UK; 3Centre for Biodiversity and Environment Research, University College London, Gower Street, London WC1E 6BT, UK; shawn.dove@hotmail.com; 4Institute of Zoology, Zoological Society of London, Outer Cir, London NW1 4SX, UK

**Keywords:** seaweed, holobiont, microbial gardening, salinity, *A. vermiculophyllum*

## Abstract

Terrestrial plants are known to “garden” the microbiota of their rhizosphere via released metabolites (that can attract beneficial microbes and deter pathogenic microbes). Such a “gardening” capacity is also known to be dynamic in plants. Although microbial “gardening” has been recently demonstrated for seaweeds, we do not know whether this capacity is a dynamic property in any aquatic flora like in terrestrial plants. Here, we tested the dynamic microbial “gardening” capacity of seaweeds using the model invasive red seaweed *Agarophyton vermiculophyllum*. Following an initial extraction of surface-associated metabolites (immediately after field collection), we conducted a long-term mesocosm experiment for 5 months to test the effect of two different salinities (low = 8.5 and medium = 16.5) on the microbial “gardening” capacity of the alga over time. We tested “gardening” capacity of *A. vermiculophyllum* originating from two different salinity levels (after 5 months treatments) in settlement assays against three disease causing pathogenic bacteria and seven protective bacteria. We also compared the capacity of the alga with field-collected samples. Abiotic factors like low salinity significantly increased the capacity of the alga to deter colonization by pathogenic bacteria while medium salinity significantly decreased the capacity of the alga over time when compared to field-collected samples. However, capacity to attract beneficial bacteria significantly decreased at both tested salinity levels when compared to field-collected samples. Dynamic microbial “gardening” capacity of a seaweed to attract beneficial bacteria and deter pathogenic bacteria is demonstrated for the first time. Such a dynamic capacity as found in the current study could also be applicable to other aquatic host–microbe interactions. Our results may provide an attractive direction of research towards manipulation of salinity and other abiotic factors leading to better defended *A. vermiculophyllum* towards pathogenic bacteria thereby enhancing sustained production of healthy *A. vermiculophyllum* in farms.

## 1. Introduction

Like all higher organisms, terrestrial plants and aquatic flora have evolved in the context of a microbial world. Decades of research have demonstrated the importance of microbes in regulating health and fitness of terrestrial plants. It is now widely acknowledged that the health, development, growth, and resistance to stressors of plants can be influenced by complex interactions with their microbial communities ultimately forming a single entity called the plant “holobiont.” Similarly, complex interactions with microbes are known to play an important role in regulating health and fitness of aquatic flora like seaweeds [1]. 

Seaweeds (or marine macroalgae) are photosynthetic autotrophs omnipresent in temperate, tropical, and Arctic marine ecosystems. They play a vital role in marine environments as foundation species and ecosystem engineers [2,3], providing a suite of ecologically valuable services like food, habitat, carbon sequestration, etc. [4,5]. Seaweeds are also highly important commercially, providing pharmaceuticals, human food, hydrocolloids, etc., with the global seaweed industry worth >USD 6 billion per annum [6]. Just like plants, seaweeds live in close association with microbes and usually need their associated bacteria to function optimally as a holobiont [1]. 

An average millilitre of seawater contains approximately 10^7^ viruses, 10^6^ bacteria, and 10^3^ microalgal cells. Thus, seaweeds are constantly exposed to microbes and live in a sea of microbes. Similar to plant surfaces that are colonized by microbes forming the rhizosphere, phyllosphere and anthosphere, seaweed surfaces are also colonized by complex microbial communities which together with the host form a unified functional entity [1]. This biofilm-like “second skin” is mainly dominated by epibacteria [7]. These microbes often play a crucial role in seaweed health, development [8], and defence against pathogenic microbes [9]. The interplay of seaweeds with bacteria affects—among other things—nutrient exchange, defence mechanisms, morphology, reproduction, and settlement. However, seaweed-bacterial interactions are not always beneficial and seaweeds are reported to suffer from a variety of disease symptoms [10,11]. Seaweeds are plagued by a variety of eukaryotic pathogens and also by equally important bacterial pathogens (reviewed by [10,11]). However, relative to terrestrial plant-microbe interactions the underlying molecular mechanisms and/or processes behind seaweed-microbe associations are rarely understood. 

A growing body of literature suggests that land plants can regulate the composition of their rhizosphere to promote the growth of microorganisms that can improve plant fitness in a given ecosystem. Released metabolites have been shown to attract beneficial microorganisms and influence the assembly of rhizosphere microbiota that enhance the capacity of plants to adapt to their environment [12]. However, while investigating seaweed host–microbe relationships, we have given little focus to the impact of such metabolites or infochemicals (information-conveying chemicals [13]) that can potentially “garden” interactions of the seaweed with microbes via exuded metabolites. 

Communication via infochemicals is mainly mediated through the seaweed surfaces, which, together with their “second skin” of microbes and an additional diffusive boundary layer, represent a microniche analogous to the rhizosphere of plant roots or the phycosphere of phytoplankton cells. Rich in infochemicals, a seaweed’s surface layer can be considered as a marketplace where cross-kingdom communications are mediated by release and uptake of metabolites. This zone has been recently defined as “eco-chemosphere” of seaweeds [14]. The diverse chemical milieu from the surface can be composed of sugars, fatty acids, amino acids, and other secondary and primary metabolites [15,16,17,18]. Infochemicals present at the surface of seaweeds have the potential to reduce or enhance colonization by microbes and other higher organisms (reviewed by [19,20]). 

Reference [21] recently demonstrated that halogenated furanones from the surface of the red seaweed *Delisea pulchra* can shape the surface microbiota of the seaweed. Additionally, an earlier work by [22] showed that surface metabolites from the brown seaweed *Fucus vesiculosus* can control biofilm formation under both laboratory and field settings. However, very little work so far has demonstrated that aquatic plants or seaweeds use surface metabolites to encourage recruitment of beneficial microbes while deterring detrimental pathogens, as is known for the rhizosphere. We recently demonstrated that the surface metabolites of the red seaweed holobiont *Agarophyton vermiculophyllum* can chemically “garden” microbial colonizers, i.e., attract settlement of beneficial microbes (protectors) that protect the alga from a tip bleaching disease symptom via a probiotic effect (attracting settlement of microbes) and, at the same time, deter settlement of bacteria (pathogens) that induce such disease via antisettlement effect [9]. 

Metabolic composition of plant root exudates is not a uniform or static property and varies depending on environmental conditions, nutrition, and soil type, among other factors [23]. Recent work by [24] on the brown alga *Lobophora rosacea* demonstrated that long-term as well as short-term exposure to low pH conditions causes significant quantitative changes to the compounds comprising the alga’s metabolome. Lobophorenols, which are known to induce allelopathic activity against corals, were found to be significantly decreased in concentration in lower pH conditions. However, this study focused on endometabolomes and not surface-associated metabolites of the alga. In addition, no empirical link was established to show how reduced lobophorenol concentration may affect ecological interactions of the alga with their competitors, i.e., corals. Thus, unlike terrestrial plants, we do not know how abiotic factors may influence the chemically mediated microbial “gardening” capacity of seaweed holobionts or any other aquatic plants, thereby influencing their capacity to manipulate interactions with beneficial and pathogenic microbes. 

Marine environments are characterized by broad fluctuations of environmental conditions, including extreme temperatures, rapid salinity and nutrient fluctuations, desiccation, intense sunlight, and others. Salinity is considered one of the most significant factors limiting the distribution of species, including seaweeds in aquatic environments [25]. The Baltic Sea is one of the largest brackish waters, with a strong estuarine-like salinity gradient that can determine the geographical distribution of many species [26]. The salinity gradient in the Baltic has also been found to structure epibacterial community compositions and their diversity on the native brown seaweed *Fucus vesiculosus* [27]. Additionally, salinity and time have been found to alter richness and beta diversity of epibacterial communities on the invasive red seaweed *A*. *vermiculophyllum* [28]. However, the driving mechanisms behind this altered community composition are still unknown. We speculate that an altered chemical composition of the algal surface metabolome over time-driven by salinity-may be the driving force behind this differential bacterial recruitment.

Thus, in the current study, we hypothesized that microbial “gardening” capacity of seaweeds are dynamic and that abiotic factors like salinity can manipulate the “gardening” capacity using the red model seaweed *A*. *vermiculophyllum*. To test this hypothesis, we conducted a 5-month mesocosm experiment, investigating the impact of two different salinities on the microbial “gardening” capacity of *A*. *vermiculophyllum*. Microbial “gardening” capacity of the alga against pathogens and protectors with field-collected samples have been published in [9]. However, to understand the effect of time, we also compared the capacity of the alga following the 5-month salinity treatments with the “gardening” capacity at the beginning of the treatment, i.e., with field-collected samples.

## 2. Materials and Methods

### 2.1. Collection of Algal Samples

*Agarophyton vermiculophyllum* (Ohmi) [29] individuals were collected from Nordstrand, Germany (54°29.166′ N, 8°48′.746′ E) in May 2015. Individuals were placed into a cooler box and transported to the laboratory within less than 2 h after they were collected. At the laboratory, they were transferred to 20 L aquaria, where they were aerated constantly and maintained at 33 psu (the approximate salinity of the collection site), 15 °C, and 75 µmol m^−2^ s^−1^ light intensity (12 h/day) for 16 h until treatment began. 

### 2.2. Experimental Setup

The experiment was conducted in a climate chamber, where two different salinity treatments (low = 8.5 ± 0.51, medium = 16.5 ± 1.44) were applied from May to October 2015 (5 months). Temperature and light intensity were maintained throughout the experiment at 15 °C [30] and 75 µmol m^−2^ s^−1^ (12 h/day), respectively. The two salinity levels, both of which fall within *A*. *vermiculophyllum*’s Baltic distribution range [31], were prepared from Kiel Fjord water (16.5 ± 0.5 psu throughout the experiment, measured weekly). Low salinity water was prepared by 1:1 fresh water (0 psu) dilution. Salt (Instant Ocean, Blacksburg, VA, USA) was then added to the low salinity water to obtain the medium salinity water. Water for both salinity levels was prepared 48 h before distribution to the individual aquaria. 

Five 8 L plastic aquaria were assigned to each salinity level, with each aquarium receiving 6 L of prepared water and an equal wet weight (ca. 60 ± 1.5 g) of *A*. *vermiculophyllum* individuals. Manual water exchange was performed three times per week for each aquarium. Wet weight of the individuals was measured once every month to monitor change in algal biomass.

### 2.3. Epibacteria

Isolation and identification of epibacterial strains used in the assays below have been previously described in [32]. For settlement tests, bacteria were precultured from cryopreserved stocks. Strains were reanimated and then maintained on MB agar medium in darkness. All cultures were incubated at 25 °C using Marine Broth for the assays.

### 2.4. Generation of Surface-Associated Metabolites

To assess the effect of salinity on chemically mediated microbial “gardening” properties of *A*. *vermiculophyllum*, surface-associated metabolites were extracted from individual *A*. *vermiculophyllum* specimens according to the method developed by [32]. Thalli of *A*. *vermiculophyllum* were submerged for 5 s in a dichloromethane-hexane solvent mix (1:4 *v*/*v*), a process which collects surface-associated metabolites while avoiding intracellular metabolites (see [32] for details). Immediately after collection from the field, surface extracts were obtained from 5 individuals and designated as T0S0, for time point T0 and field salinity S0 (i.e., 33 psu). Defence capacity against pathogen and protectors at T0S0 have been published in [9] and serve here as reference samples to compare the capacity of the alga at T1S1 and T1S2 with field-collected samples. 

After 5 months of salinity and time treatment, surface extracts were collected from 5 individuals from each of the low and medium salinity levels, i.e., T1S1 and T1S2, respectively. The prepared 15 extracts (from T0S0, T1S1, and T1S2) containing the surface-associated metabolites were filtered through GF/A filter paper (Whatman Ø = 15 mm) to remove particles, and the solvent was evaporated under a vacuum at 20 °C, using a rotary evaporator (Laborota 3000, Heidolph, Schwabach, Germany). The upconcentrated extract was then taken up in acetonitrile in such a way that 1.5 µL of the upconcentrated extract contained the same concentration of metabolites as extracted from an algal surface of 99.64 mm^2^ [9]. 

### 2.5. Defence Capacity Test of A. vermiculophyllum Surface Metabolites against Pathogens, Protectors, and Reference Fouling Bacteria

Inhibition of or reduction in bacterial settlement and attachment represents the first line of defence against microbial challenge [33]. Thus, antisettlement assays were conducted as the most relevant criterion for testing antimicrofouling defence because it quantifies both repellent and toxic effects [34]. The assay was performed according to the method described in [9,32]. In total, 15 extracts were tested (see above). The extract was then used to coat each replicate well with a surface area of 99.64 mm^2^ (using a 96 well plate). Each extract was subreplicated 3 times to account for variability in bacterial settlement response. Acetonitrile was then evaporated and metabolites originating from the surface of the alga remained on the surface of the well, allowing us to test at an ecologically realistic 1-fold concentration. Solvent residue (without *A*. *vermiculophyllum* extract) was used as control wells. 

To account for the variability in the bacterial settlement rates, each *A*. *vermiculophyllum* extract was subdivided and tested in three pseudo replicates against each bacterial isolate. Results obtained for pseudo replicates were averaged before statistical analyses were conducted. The tested target strains were selected based on results from the tip bleaching assay described in [9] where pathogens are the bacterial isolates that induced risk of tip bleaching and protectors are the strains that reduced the risk of tip bleaching. All three pathogens (*Kordia algicida*, *Croceitalea eckloniae*, and *Pseudoalteromonas arctica*—all three isolated from invasive range) were tested in the antisettlement assays. In addition, seven protectors, i.e., *Ralstonia sp., Tenacibaculum skagerrakense, Tenacibaculum aestuarii, Alteromonas stellipolaris,* and *Nonlabens dokdonensis*—all five strains isolated from invasive range, and *Shewanella aquimarina* and *Cobetia marina*—both strains isolated from native range, were tested. To investigate any shift in generic defence capacity of the alga, surface extracts generated from T0S0, T1S1, and T1S2 were also tested against a set of 11 reference marine strains isolated from stones located near the alga and seawater. These strains were: *Vibrio sp*., *Pseudoalteromonas sp*., *Alteromonas sp*., *Winogradskyella sp*., *Loktanella sp*., *Psychroserpens sp*., *Bacillus amyloliqueifaciens*, *Aquimarina sp*., *Maribacter sp*., *Bacillus aquimaris*, and *Cytophaga sp*. 

Suspensions of 106 µL of each of these bacterial strains (O.D. 0.6–0.8), after culturing in MB liquid medium (as described above), were pipetted into wells. After letting the bacteria settle for 3 h, the wells were rinsed twice with 110 µL sterile seawater to remove unattached cells. Cells that remained after rinsing were stained with SYTO 9 (Invitrogen, GmbH), a fluorescent dye that binds DNA. Fluorescence measured (excitation, 477–491 nm; emission, 540 nm) by a plate reader was used as an estimate of the cell density of settled bacteria. Settlement was tested for all of the above-mentioned bacterial strains on each extract. 

### 2.6. Statistical Analysis

“Log effect ratio” was used to represent the defence strength of *A*. *vermiculophyllum* surface metabolites. It is calculated by dividing the logarithm of the fluorescence measured for the settled bacteria of strain Y when surface metabolites are present by the fluorescence of the same strain when surface metabolites are absent. If the fluorescence values are equal (the number of settled bacteria is the same whether or not surface metabolites are present), the log effect ratio value will be 0, indicating that the surface metabolites being tested do not have an effect on settlement. The log effect ratio value will be negative if the surface metabolites have a deterrent effect or positive if there is an attractant effect. A value of −1 would mean the surface metabolites cause a 10-fold reduction in bacterial settlement, whereas +1 would mean a 10-fold enhancement. A one-way ANOVA was conducted to compare the log effect ratios of defensive strength between samples prior to the start of the experiment, i.e., reference samples (T0) with field salinity (S0), i.e., T0S0; samples generated after 5 months (T1) of low salinity treatment (S1), i.e., T1S1; and samples generated after 5 months (T1) of medium salinity treatment (S2), i.e., T1S2. Post hoc comparisons were conducted using Tukey’s honestly significant difference test (HSD) (*p* = 0.05). Prior to all analyses, the data (Appendix A) were tested for normality and homogeneity of variances with the Shapiro–Wilk and Levene’s tests, respectively. Statistica software (Stat Soft, Tulsa, OK, USA) was used to conduct statistical tests..
(1)$$\mathrm{Defence}\;\mathrm{strength}=\log\frac{\left(\mathrm{bacterial}\;\mathrm{settlement}\;\mathrm{in}\;\mathrm{presence}\;\mathrm{of}\;\mathrm{surface}\;\mathrm{metabolites}\;\mathrm{of}\;A.\;vermiculophyllum\;\right)\;}{\left(\mathrm{bacterial}\;\mathrm{settlement}\;\mathrm{in}\;\mathrm{absence}\;\mathrm{of}\;\mathrm{surface}\;\mathrm{metabolites}\;\mathrm{of}\;A.\;vermiculophyllum\right)}.$$

## 3. Results

### Defence Capacity Test of A. vermiculophyllum Surface Metabolites against (i) Pathogens, (ii) Protectors, and (iii) Reference Fouling Bacteria

The effect of *A*. *vermiculophyllum* surface-associated metabolites on the settlement of pathogens differed significantly between the low and medium salinity and also when compared to reference samples (Figure 1i, 1-way ANOVA, *p* < 0.0001). The mean extract activity strength of samples from T1S1 (the lower salinity level after 5 months treatment) against pathogen settlers was significantly increased when compared to field-collected samples, i.e., T0S0. However, for samples generated from T1S2 (the medium salinity level after 5 months treatment), the activity strength was found to be significantly lower when compared to T0S0 and T1S1 samples. 

The prosettlement (or probiotic) effect of *A*. *vermiculophyllum* surface-associated metabolites did not vary significantly between samples generated from low and medium salinity levels after 5 months of treatment, i.e., T1S1 and T1S2, respectively. However, the effect was significantly higher in field-collected samples, i.e., T0S0 when compared to both T1S1 and T1S2 samples (Figure 1ii, 1-way ANOVA, *p* < 0.0001). 

With respect to the 11 reference strains tested, the activity strength of *A*. *vermiculophyllum* extracts was significantly higher in T1S1 than in field-collected samples (i.e., T0S0), but not significantly different from T1S2 samples (Figure 1iii, 1-way ANOVA, *p* < 0.0001).

## 4. Discussion

Secretion of defence compounds into the rhizosphere of terrestrial plant roots has been found to be a tightly controlled, temporally dynamic process. The data presented here demonstrate for the first time that chemically mediated microbial “gardening” capacity of aquatic macrophytes like seaweeds is also not a static property, but strikingly dynamic. Abiotic factors like salinity can increase the defence capacity of the algal holobiont to deter colonization by pathogens over time, as observed for the low salinity treatment. At the same time, an increase in salinity from low (8.5, salinity in the Baltic proper) to medium (16.5, average salinity of the Kiel Fjord) resulted in a reduction in this capacity to deter pathogens. Interestingly, the probiotic effect of the surface-associated metabolites (surface metabolome) towards beneficial bacteria was found to be significantly lower in both salinity levels treated over time when compared to the field-collected samples (33, salinity of the North Sea during collection). Thus, the alga significantly lost its capacity to attract beneficial bacteria over time when compared to field-collected material. 

A potential factor that may contribute to this differential effect of surface-associated metabolites towards pathogens and beneficial bacteria might be a change in the physiological status of the alga, which is known to cause changes in exuded carbon, thereby changing the surface chemistry of the alga [35,36]. We monitored the change in the biomass of the alga between the start and end of the experiment. Although the experiment started with similar biomass of the alga at both salinity levels, at the end of the experiment, there was a significant increase in biomass of the lower salinity level alga compared with the medium salinity level (Appendix A), indicating a better physiological status at the lower salinity level.

Surface-associated metabolites can be produced by either the host alga or the microbiota individually, by both, and as a product via the induction of signalling compounds produced by either partner of the holobiont. Thus, in microbial chemical ecology, it is a major challenge to identify the origin of these infochemicals. Transcriptomic analysis of the red seaweed *Laurencia dendroidea* and its microbiome demonstrated biosynthesis of terpenoids through the mevalonate-independent pathway by the seaweed itself [35,36,37]. By contrast, DMSP (a known antibacterial compound from the surface of the brown seaweed *Fucus vesiculosus* [36]) can be contributed to the surface both by the algal host and the surface-associated bacteria, as marine bacteria are known to be DMSP producers [38,39]. Contribution towards the chemical cocktail will depend on the abundance and activity of epibacteria in the “second skin” and their metabolic activity [40]. Epibacterial communities from the low and medium salinity levels (harvested at the end of the experiment, i.e., October) using the same individuals showed a significant variation in community composition among low and medium salinity levels [28]. Community composition also changed over time, with significantly different epibacterial communities between field-collected (May) and low and medium salinity samples (collected in October). Thus, an alteration in the bacterial community composition under low and medium salinity treatments (when compared to field collected samples) may also have resulted in a difference in the cocktail of surface-associated metabolites driving change of the surface metabolome of the algal holobiont. For the *A*. *vermiculophyllum* holobiont, we do not yet know the identity of bioactive surface-associated active metabolites. Thus, we cannot discriminate the source of the surface metabolites, i.e., whether they originate from the algal host *A*. *vermiculophyllum* and/or from surface-associated microbiota. Additionally, due to shortage of generated surface extract, it was not possible for us to run further chemical analysis using metabolomics approaches and investigate which compounds were up- or downregulated among the two salinity treatments and field-collected samples that resulted in a dynamic microbial “gardening” capacity of the alga.

Plants are constantly exposed to a large variety of natural microbial enemies that can trigger infection. To counteract infection, plants are well known to release root exudates consisting of constitutively secreted and inducible phytochemicals that can directly repel, inhibit, or kill pathogenic microbes in the rhizosphere [41]. In addition to repelling pathogens, root exudates are also known to mediate interactions with beneficial microbes and engage in tripartite interactions [42]. Similar to plant surfaces, infochemicals from surfaces of seaweeds are also known to have a multitude of functions in ecological interactions with the microbial communities, e.g., by acting not only as signalling molecules, attractants, or stimulants but also inhibitors or repellents towards microbes and other colonizers like mussels, barnacles, and epiphytic algae [19,20]. These surface-associated metabolites are also known to play a strong active role in selecting epibacterial communities [22,36,43,44]. However, the function of these recruited epibacteria was not demonstrated in earlier studies. Interestingly, following a community composition analysis, protective bacteria such as *Nonlabens dokdonensis* and *Tenacibaculum sp*. (both tested in the current study) were found to be present in significantly higher abundance on surfaces of the alga when compared to the bacterial community of the tank wall [28]. Protective bacteria belonging to the genus *Alteromonas* and *Tenacibaculum* were also significantly present at higher abundance in the medium compared to the low salinity level alga’s surface. In our recent work, we found that these protective bacteria offer associational defence to the alga by reducing the risk of tip bleaching disease [9]. Given that defence against pathogens was found to be significantly higher in low salinity treated alga compared to medium salinity treated alga, it may be possible that the alga had to upregulate its defence against pathogens due to lesser abundance of protective bacteria (and thereby less associational defence provided by the protectors) when compared to the medium salinity level alga. Possible higher abundance of pathogens compared to medium salinity treatment might also have led to upregulation of defence capacity of the alga against pathogens. However, we have no information regarding abundance of the pathogens (tested in the antisettlement assays) at both salinity levels. 

Environmental factors like light, temperature, and soil moisture are known to alter the blend of chemicals in root exudates of plants. Production of defence metabolites in seaweeds is generally a plastic trait that can be influenced by both biotic and abiotic factors (e.g., [35,36,45]). Additionally, climate change stressors like ocean acidification have been found to alter the endometabolome of seaweeds (e.g., [46]). Adaptation in chemical defence towards common bacterial foulers (bacteria that colonize surfaces of artificial substrates) has been previously reported for *A*. *vermiculophyllum* (but over a longer time scale, i.e., 10 years) when invasive populations of the alga were found to have adapted their defence capacity to cope with the new bacterial foulers in the invaded range but had lost the capacity to fend off old foulers from the native range [32]. In the present study, the defence capacity of the alga towards pathogenic epibacteria changed with salinity and over time. The probiotic effect towards beneficial bacteria also changed over time but did not vary significantly between low and medium salinity treated alga. This suggests that the surface metabolome of the alga has undergone a change due to the treatment and over a temporal scale, causing a differential anti- and probiotic effect. 

Soil salinity is a major abiotic stress in land plant agriculture worldwide. Salt stress is also one of the most important environmental stress factors that can affect wetland plants and has been found to have a significant effect on the total amino acid content from root exudates of wetland plants [47]. A few studies have shown that phlorotannin contents in brown macroalgae can increase with increasing salinity [48,49]. Production of the defence compound sesquiterpene elatol by *Laurencia dendroidea* was found to be at its maximum at 35, the optimal salinity condition for growth of the alga. However, elatol production declined at a severe salinity of 40 psu [35]. In this light, it is not unlikely that changes in defence-related surface metabolomes of *A*. *vermiculophyllum* may be directly due to salinity stress or acclimation to salinity change, resulting in significantly altered interactions with beneficial and detrimental bacteria over time. 

Defence capacity against 11 tested reference strains was significantly increased over time for low salinity treatments when compared to field-collected samples but was not significantly different among the two salinity treatments. Functions of these reference fouling strains are not known but they were isolated from stones lying adjacent to the alga. A decrease in salinity from 33 (field-collected samples) to 8.5 caused an enhancement in defence capacity against this pool of settlers over time, indicating that a reduction in salinity and thus a low saline environment might be beneficial for the alga in terms of preventing surface colonization. Surface-associated chemical defence of seaweeds (against micro- and macrocolonizers), including *A*. *vermiculophyllum*, is known to undergo seasonal fluctuations, indicating a change in the surface chemistry of the alga over time [50,51]. A recent study by [43] showed that surface-associated metabolites of the Mediterranean brown seaweed *Taonia atomaria* can change at a temporal scale over a year in response to environmental factors and host physiology. This change in surface metabolome over season was shown to play a role in structuring the surface bacterial microbiota of *T*. *atomaria*. 

## 5. Conclusions

In conclusion, our study demonstrates that microbial “gardening” capacity of seaweed is a dynamic property, as in terrestrial plants. Variability in the anti- and probiotic effect of the surface-associated metabolites towards pathogenic and protective bacteria was observed due to changes in salinity and additionally at a temporal scale. Paix et al., 2020 [38] demonstrated that surface-associated metabolites of *T*. *atomaria* can drive the zonal variability of epibacterial communities at a thallus scale. *A*. *vermiculophyllum* is known to exhibit seasonal variation in its surface defence chemistry towards epiphytic colonizers [51]. Although in this current study we have not investigated the surface metabolome of *A*. *vermiculophyllum* at different timepoints, i.e., start (T0) and end (T1) of the experiment, we may speculate that a seasonal fluctuation in the surface metabolome of the alga may have resulted in the selective recruitment of the beneficial and detrimental bacteria on the surface of the alga and thereby differential anti- and probiotic capacity of the alga at a temporal scale. However, this hypothesis needs to be tested in the future. Identification of up- and downregulated surface metabolites through untargeted metabolomics in future may allow us to identify the compounds responsible for such differential microbial “gardening” capacity—providing a mechanistic underpinning and an attractive direction for the manipulation of abiotic factors like salinity and thereby the anti- and probiotic effect of the alga for beneficial outcomes like disease prevention and sustained production in *A*. *vermiculophyllum* farming.

## Figures and Tables

**Figure 1 microorganisms-08-01893-f001:**
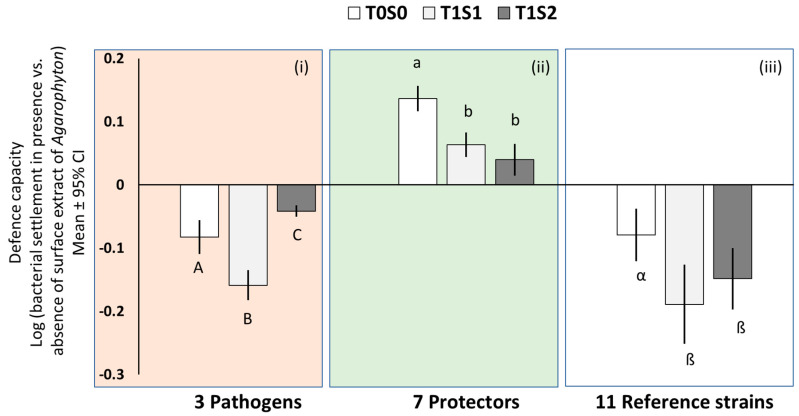
Mean antisettlement activity of *A*. *vermiculophyllum* surface-associated metabolites originating from field-collected samples (T0S0), samples generated after 5 months of low salinity treatment (T1S1), and samples generated after 5 months of medium salinity treatment (T1S2) against 3 pathogenic (**i**; 1-way ANOVA, Tukey’s honestly significant difference test (HSD), *p* < 0.0001), 7 protective (**ii**; 1-way ANOVA, Tukey’s HSD, *p* < 0.0001), and 11 reference strains (**iii**; 1-way ANOVA, Tukey’s HSD, *p* < 0.0001). Capital Latin letters indicate extracts that significantly differed in defence activity against pathogens, whereas small Latin letters indicate extracts that significantly differed in probiotic activity against protectors. Greek letters indicate extracts with significantly different activity against tested reference strains. Error bars ± CI (*n* = 5). **Note:** Defence capacity (against pathogen and protectors only) at T0S0 have been published in [9] and serve here as reference time points to compare the capacity of the alga at T1S1 with T0S0 and T1S2 with T0S0.

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
