# Peer review of "Chemically Mediated Microbial “Gardening” Capacity of a Seaweed Holobiont Is Dynamic"

_microorganisms, 2020, doi:10.3390/microorganisms8121893_

Round 1
Reviewer 1 Report
This paper has some interesting points to discuss but I would suggest to consider some questions to be published.
Introduction and discussion are too long with several mentions to higher plants which are unnecessary in the paper.
In Material and Methods, in the heading on “isolation , identification,…” , authors write that isolation and identification of bacteria were previously described by Saha et al. However they do not show that bacteria were obtained at any time.
They have also mentioned papers previously published by one of authors of this ms submitted to revision. I think too much information is supported on those previously papers but little new one. For instance, some examples:
Pg 4, line 158-159
Isolation and identification of bacterial strains have been previously described in Saha et al. 158 (2016) and Saha and Weinberger (2019). No new data
Pg 4, line 170-171
Individual colonies were picked and cultured again on MB agar plates 169 until a pure culture was obtained. Identification of strains was done using 16S rRNA sequencing 170 (described in Saha, Wiese, Weinberger, and Wahl, 2016). The Cryobank System (Mast Diagnostica 171 GmbH, Reinfeld, Germany) was used to preserve the identified strains at -80°C, following the manufacturer's instructions, and they were later tested in the bioassays described below. It seems that was already carried out in Saha et al. 158 (2016) and Saha and Weinberger (2019)
Pg 5, line 200
The assay was performed according to Saha et al. (2016) and Saha and 200 Weinberger (2019). what is then the difference with the submitted ms?
Pg 6, line 247-248
Results. Irrespective of the salinity treatments all Agarophyton extracts deterred pathogens and reference strains but attracted protective microorganisms as observed in our previous study (Saha and Weinberger 2019).
Results
Please avoid to mix results with discussion.
Authors write on probiotic effect and 11 reference strains but neither probiotic effect has been described in introduction and M&M nor 11 reference strains are showed.
Also they speak on pathogens and protectors but this reader do not know what organisms comprise these classification. What strains are pathogens and what protectors? are bacterial strains from Korean samples included in this classification too?
Discussion is a struggle for this reviewer.
As said before, discussion is tangled. Newly mentions to higher plants are done but not necessary. Focus no seaweeds or algae!!
In page 7 (lines 295), authors say that algal biomass was monitored , but any mention in M&M is showed (?)
The presence of DMSP is not completely understood here . At least in the form that has been introduced.
More things:
Pag 7
Epibacterial communities: what strains compose this community?
Other example:
Protective bacteria belonging to the genus Alteromonas and Tenacibaculum were also significantly present at higher abundance in the medium compared to the low salinity level alga’s surface. In our recent work, we found that these protective bacteria offer associational defence to the alga by reducing the risk of tip bleaching disease (Saha and Weinberger 2019).
What bacteria comprise the protective bacteria in this ms??
The protective bacteria are the same in this work or were isolated from Korean samples…
I think discussion should focus on the discussion of results.
References are not consistent (e..g. [33], Saha et al. 2014, Saha, Wiese, Weinberger, and Wahl, 2016).
Reviewer 2 Report
Dear Authors,
Your work is an interesting research work on the seaweeds’ biofilm “gardening”, as you say. You prove that environmental parameters (salinity over time) affect the quantity and quality of the microbes present in the surface of the seaweeds. These microbes interfere with surface-associated metabolites, which affect their anti- and probiotic effects.
Yet, because the experimental part of the paper is very simple (only one set of experiments were performed) and it seems part of a broader experiment. I believe that this should be a short paper.
Simple and clear experimental design. Clear results were obtained.
The introduction describes extensively the state of the art.
Besides, the manuscript needs to be improved for the discussion is speculative and makes statements that have not been addressed in this experimental part.
Finally, the references are not in accordance with the editorial rules, and several references are cited, neglecting the recommended way to quote them.
Details are shown bellow.
Specific remarks.
Abstract
Although it is advisable the abstract has detailed information, some information is not fundamental. Also, it is not in the proper order: first the methods, then the results, then the conclusions. I suggest reviewing the abstract.
Line 14 – I wouldn’t say seaweeds are macrophytes: I use this name exclusively for vascular plants. Could you clarify this?
Line 36 and onwards – Agarophyton must be written in italic. The correct form to write the species is, in fact, A. vermiculophyllum.
Throughout the abstract, you use both seaweeds and macrophytes as synonyms but they are not.
Introduction
Line 48 – same remark made for the abstract regarding macrophytes.
Line 49 – Brown seaweeds also thrive in cold waters.
Line 51 – the correct way to cite is, I believe, [2,3] and [4,5].
Line 61 – wrong way citation (Wichard et al., 2015) – should be [8].
Line 62 - wrong way citation (Saha and Weinberger, 2019) – should be [9], not [10].
Line 65 – the correct way to cite is, I believe, [11,12]
Line 67 - wrong way citation (Gachon et al. 2010; Egan et al. 2014) – should be [11,12].
Line 71, 82, 90, 92, 103, 105, 124, 149, … - wrong way citation. Also, after citing an author such as in line 78, Voges et al. 2019 add the number of the reference “Voges et al. 2019 [15] recently […]”. Make a full revision of the paper.
Line 89 – and OTHER secondary and primary metabolites.
Line 124 - Saha et al. 2020 is not in the references.
Line 129 substitute macrophytes by seaweeds.
Material and methods
Experimental setup
- In what type of substrate was vermiculophyllum growing? If it is muddy (as usual) a lot of debris was transported with the biomass. So didn’t you wash your biomass with clean water to remove these contaminants?
- Why did you choose 15ºC? It seems a rather low temperature for this species.
- Did you provide nutrient media to the seaweeds?
- So you used a 10g/L biomass, which is too high for this species to thrive in aquaculture. I now you didn’t mean to make it grow, but with this high biomass density in each aquarium, you probably had negative interference between seaweed specimens and their microbiota.
Isolation, identification and culture of epibacterial strains
Line 180 – substitute “branches” by “thalli”, the correct word when you are referring to simple pluricellular organisms such as seaweeds.
Line 183 - Do you mean 33 psu or 33 extracts? If you mean 33 extracts what is the original salinity of the site?
Discussion
A general remark is that you keep comparing plants to seaweeds. Brown seaweeds are not plants – they are phylogenetically very distant from plants. Besides vascular plants are much more complex and produce compounds that are not present (as far as we know) in simpler photosynthetic organisms. Thus, these comparisons must be made with caution. I advise you to exclude such references and use only those referring to seaweeds.
Besides, this has been a very narrow experiment: you have no data on the chemical composition of the biofilm nor or the microbiota taxa. For e.g. you refer in lines 338 and 339 that “Protective bacteria belonging to the genus Alteromonas and Tenacibaculum were also significantly present at higher abundance in the medium compared to the low salinity level alga’s surface” but you present no data on the abundance of any of the microorganisms tested, so you can not state that these species (or other epibionts) were present and that their abundance changed during the experiment. You can only say that their defense capacity changed and that this is due to unknown metabolites.
So, everything you wrote between lines 341 to 347 is speculative.
Line 358 - I suggest deleting the sentence “Functions of these bacteria were not tested in the previous study.” . You make a lot of comparisons with your data form another paper, previously published, but this should be an independent work, otherwise, these data should have been published together.
Line 387 and following – again speculative sentence. You presented no data whatsoever on the matter, so you shouldn’t speculate.
Conclusions
Line 394 –You say that reference [38] shows that there are seasonal changes in the metabolome. So physicochemical parameters are interfering with the epibiota. So you demonstrate, but not for the first time. I suggest removing the “for the first time”.
References
References [8] and [9] are repeated but are a bit different from each other. Correct it.
Round 2
Reviewer 1 Report
I have revised the manuscript “Chemically mediated microbial “gardening” capacity 2 of a seaweed holobiont is dynamic" by Saha et al. and I think it has greatly improved its reading with the new changes and corrections made in the manuscript.
Even though authors tried their best to justify the heading of isolation nd identification,...I would have expected that authors showed identified bacteria through 16s RNA (e..g. as supplementary information o clarifying and shortening this epigraph). So, are all identified bacteria in [31] mentioned in this paper? Otherwise if as authors say, this information was previously published in [31], then relative information in this paper should be summarized.
The rest of the manuscript seems to me well worked.
Anyway I understand that authors han made effort to improve this ms
Kind regards and good luck.
Author Response
This has now been revised, please see line 150-163. Thanks.
Reviewer 2 Report
The authors answered the questions rigorously, corrected all errors detected, and made all suggested changes. Thus, I understand that the manuscript is in ready to be published in this form.
A correction only, in the conclusions, the "aquatic macrophytes" should be corrected to "seaweeds".
Author Response
This has been now revised as suggested. Thanks.